



# Investigating rough single fracture permeabilities with persistent homology

Marco Fuchs[1,*], Anna Suzuki[2], Togo Hasumi[2], Philipp Blum[1]

[1] Institute of Applied Geosciences (AGW), Karlsruhe Institute of Technology (KIT), Kaiserstraße 12, 76131 Karlsruhe, Germany

[2] Institute of Fluid Science, Tohoku University, 2-1-1 Katahira, Aoba-ku, Sendai, Miyagi, 980-8577, Japan

* Corresponding author: Marco Fuchs, Email: marco.fuchs@kit.edu

## Abstract

The permeability of rock fractures is a crucial parameter for flow processes in the subsurface. In the last decades different methods were developed to investigate on permeability in fractures, such as flow through experiments, numerical flow simulations or empirical equations. In recent years, the topological method persistent homology was also used to estimate permeability in fracture networks and porous rocks, but not for rough single fractures. Hence, we apply persistent homology analysis on a decimeter-scale, rough sandstone bedding joint. To investigate on the influence of roughness, three different data sets are created to perform the analysis: (1) 200 µm resolution, (2) 100 µm resolution and (3) 50 µm resolution. All estimated permeabilities were then compared to values derived by experimental air permeameter measurements and numerical flow simulation. The results reveal that persistent homology analysis is able to estimate the permeability of a single fracture even if it tends to slightly overestimate permeabilities compared to conventional methods. Previous studies using porous media showed the same overestimation trend. Furthermore, expenditure of time for persistent homology analysis as well as air permeameter measurements and numerical flow simulation was compared which showed that persistent homology analysis can be also an acceptable alternative for conventional methods in this regard.

## 1    Introduction

The permeability of rocks is a crucial parameter for fluid flow processes in the subsurface. While the prevailing flow processes in porous media are well understood, a different picture emerges when the flow is dominated by fractures (Suzuki et al., 2019).

Although flow in geological settings controlled by fractures is occurring in both shallow aquifers (e.g. for drinking water supply) and deep reservoirs (e.g. geothermal energy production, oil and gas abstraction), fractures are often simplified as two parallel plates. In addition, due to complexity reasons, a single parameter, the hydraulic aperture, is often used to represent the permeability of a single fracture or even entire discrete fracture networks (Min et al., 2004; Blum et al., 2009; Müller et al., 2010). However, due to roughness of fracture surfaces, a single value is not sufficient to capture the flow channeling and

critical flow paths (Tsang, 1992; Tsang and Neretnieks, 1998). Hence, investigations of the fracture roughness are essential, although this is more expensive in terms of costs and time (Tatone and Grasselli, 2012, 2013).

Nowadays, various methods are available to study the flow influenced by fracture roughness through single fractures. These can be divided into four major categories: (1) empirical methods, (2) experimental methods, (3) numerical methods, and (4) geometric methods.

Empirical methods are simple but also fast, cheap and often sufficient to provide a first overview over the flow behavior of a fracture. There are different empirical models derived from flow experiments, numerical simulations or statistical models of different fracture types (Louis, 1972; Barton and Quadros, 1997; Xiong et al., 2011; Kling et al., 2017; Suzuki et al., 2017). Often, solely mechanical apertures and relative roughness depending on the standard deviation are required to calculate hydraulic apertures but also fractal dimension or Peklenik number defined as the ratio of the correlation length in x- and y-

direction (Patir and Cheng, 1978; Brown, 1987).

Experimental methods provide more detailed and scientifically based results than empirical methods. The standard methodology is flow-through experiments in laboratory scale to observe flow patterns in single fractures (Brown et al., 1998; Watanabe et al., 2008; Ferer et al., 2011). In recent years, hydro-mechanical coupling (Vogler et al., 2018; Wang et al., 2021) or reactive transport are additionally performed to the exclusive investigation of flow patterns in flow-through experiments

(Durham et al., 2001; Huerta et al., 2013). In addition to typical flow tests on laboratory scale, it is also possible to perform flow experiments on larger scales in the laboratory or in the field (Novakowski and Lapcevic, 1994; Thörn and Fransson, 2015;

Weede and Hötzl, 2005). Although, flow-through experiments are a method to investigate flow in fractures directly, it is just possible to predict the flow distribution and preferred flow path within the fracture, if the geometry is replicated with transparent materials. Beside flow-through experiments, air permeameters can be used to determine permeability of fractures

(Cheng et al., 2020; Hale et al., 2020; Hale and Blum, 2022). An air permeameter allows to measure permeability of fractures directly on an outcrop or drilling core (Brown and Smith, 2013). In addition, it is also possible to obtain a zonal observation of the permeability, since several measurements have to be conducted along an edge due to the measurement method (Hale et al., 2020).

Another way to investigate flow in fractures is the representation of the fracture in numerical models. For this purpose, the

geometry of a fracture is either projected onto a two-dimensional surface (Pyrak-Nolte and Morris, 2000; Javanmard et al., 2021) or represented in three dimensions (Javadi et al., 2010; Xiong et al., 2011; Wang et al., 2016; Chen et al., 2021). The latter increases the computational effort considerably, but also allows a more accurate investigation of flow processes. Another major advantage is that it is possible to simulate various scenarios under different conditions, such as high confining pressures or high flow rates, which is not possible to reconstruct in a laboratory experiment. In addition, numerical models are able to

consider flow effects region by region in the fracture and therefore characterize main flow paths (Marchand et al., 2020; Javanmard et al., 2021). However, it has to be considered that the geometry of the fracture and the prevailing boundary conditions for simulations have to be precisely known in order to obtain meaningful results (Barton et al., 1985; Tatone and Grasselli, 2012).

However, if the focus is set on the investigation of the flow behavior and permeability distribution within the fracture,

geometric methods can also be used. The Kozeny-Carman equation is well-known as a representative method for estimating flow properties from pore structures (Kozeny, 1927; Carman, 1937), and attempts have long been made to estimate permeability by extracting porosity from images without experiment or numerical simulation (Costa, 2006; Torskaya et al., 2014; Oliveira et al., 2020). More recently, attempts have been made to use machine learning or deep learning on images (Sudakov et al., 2019; Anderson et al., 2020; Araya-Polo et al., 2020; Hong and Liu, 2020; Alqahtani et al., 2021; Da Wang

et al., 2021). Of course, deep learning is a powerful method, but it has the problem that its contents become a black box and also it is dependent on the training data.

Topological data analysis (TDA) is another way to extract crucial information of shapes and structures in big data (Carlsson, 2009; Thiele et al., 2016). TDA is an analysis method that focuses on the structure of data based on the field of algebraic topology and has strengths in data such as images, complex structures, and networks. TDA can capture the structure of the

data in a rough sense and obtain qualitative features, therefore, is robust to noise and independent of coordinate system and number of dimensions. Persistent homology, one of the most leading TDAs, can characterize structures of data by capturing how holes appear and disappear. This method was already developed in the early 2000s (Edelsbrunner et al., 2000; Zomorodian and Carlsson, 2005) and is applied in various research fields such as materials science (Hiraoka et al., 2016), computer science (Choudhury et al., 2012) or biology (Chan et al., 2013). In geosciences, it has only been applied in the last decade with typical

applications in the characterization of porous rocks and the determination of the permeability of such (Delgado-Friedrichs et al., 2014; Robins et al., 2016; Bizhani and Haeri Ardakani, 2021). Furthermore, the determination of hydro-elastic properties of porous media is possible with this method (Jiang et al., 2018). In the field of fractured rocks, persistent homology (PH) was recently also applied to study small-scale fracture networks (Suzuki et al., 2020; Suzuki et al., 2021). Based on these studies, the general application of PH for permeability estimation of fracture networks could be demonstrated. In these small-scale

(millimeter to centimeter scale) studies the effect of fracture roughness was not crucial for flow behavior or was not particularly investigated. Further research is therefore needed to investigate larger-scale fractured rocks, in which surface roughness has a significant effect on flow behavior.

Hence, the objective of this study is the application of the persistent homology analysis on a natural, mesoscale (decimeter scale) single fracture to estimate the permeability. The focus is on the influence of roughness of the fracture surfaces on the

flow behavior and the determination of the permeability distribution across a natural bedding plane fracture. In order to additionally investigate the influence of resolution, three data set of the same fracture are prepared, which have different resolutions (50 µm, 100 µm and 200 µm). Finally, these results are compared with results from experimental air permeameter measurements as well as numerical flow simulations.



## 2    Methods

### 2.1    Fracture Sample

The fracture sample is a natural bedding joint in a sandstone block taken from a quarry in Bebertal, Germany (Figure 1; Heidsiek et al., 2020; Hale and Blum, 2022). The sandstone is Flechtinger sandstone, an oil and gas reservoir rock in the Northern German Basin. The block contains one bedding joint with a rough extent of 120 mm in x-direction and 450 mm in y-direction (Figure 1). Previous studies already characterized important hydro-mechanical parameters such as porosity, matrix and fracture permeability, Young's and bulk modulus, thermal dependencies of stress and strain behavior and the mineralogical composition (Frank et al., 2020; Cheng et al., 2020; Fischer et al., 2012; Hale and Blum, 2022; Heidsiek et al., 2020; Hale et al., 2020; Hassanzadegan et al., 2012; Blöcher et al., 2019). Of particular interest for this study is the low matrix permeability of 0.1-10 mD, which allows it to be considered almost impermeable (Cheng et al., 2020; Hassanzadegan et al., 2012). Furthermore, the findings of Hale et al. (2020) and Hale and Blum (2022) are seminal, since they performed investigations on fracture permeability on exactly the same fracture.

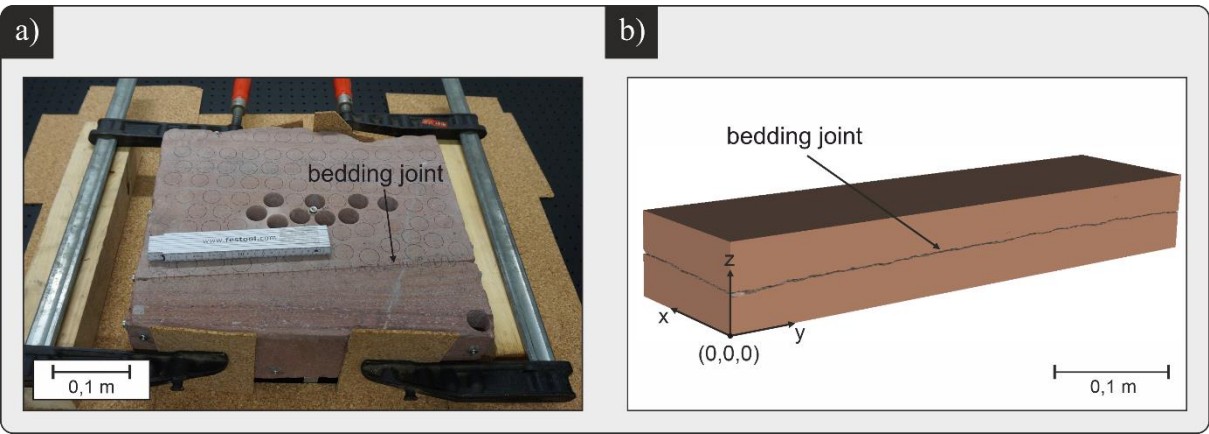

Figure 1: a) Photo of the studied sandstone block showing also the investigated bedding joint. The front surface of the real block corresponds to the y-z-plane in the digital model on the right side. b) 3D model of the bedding joint and the surrounding sandstone block.

### 2.2    Matching and Binary Image Generation

For data preparation, a self-developed Python code called "MatchPy" was used. This code is able to (1) match two separated and spatially uncorrelated fracture surfaces and (2) create binary cross-sections of the fracture as input for the PH analysis.





In the matching process, meshed laser scans of the bedding joint surfaces were used as input data for the Python code (Figure 2). The surfaces were scanned by a combined system of the high-resolution laser scanner Nikon ModelMaker MMDx100 and

the articulated arm MCA II, on which the scanner was mounted (Nikon Metrology NV, 2010, 2018). The scanner provides a resolution of 100 µm and a single-point-accuracy of 10 µm (Nikon Metrology NV, 2018). The scanned point cloud was then meshed using MeshLab (Cignoni et al., 2008). Since the meshes were not spatially related, the two surfaces were roughly matched by hand to shorten the runtime of the Python code for data preparation. The exact matching was then performed stepwise by several rotation and translation steps within specified limits using the Python code. The translation limit was 3 mm

total distance in each direction, in which one surface was displaced in 100 µm steps, starting from the geometric center of the surface. In addition, there was a rotation range of -0.3° to 0.3°, in which rotation was performed in 0.01° steps. The best fit was then determined applying a minimization function of the average mechanical aperture using the arithmetic mean between the fracture surfaces.





Figure 2: Workflow of this study with three distinct steps, which is applied on the laser scanned fracture surfaces.



## 2.3 Persistent Homology

The software HomCloud was used to analyze image data based on persistent homology (PH) (Obayashi et al., 2022). HomCloud allows us to extract flow-channel information from black-and-white image data (Suzuki et al., 2021). The first step is to convert the fracture information prepared in Section 2.2 into a 3D binarized image datasets.

Three-dimensional image construction of the bedding joint and the surrounding sandstone block was generated. The image construction is a series of binary cross-sectional images of the xz planes along the y-axis. The size of the image area is 115 mm × 8.4 mm × 3,922 mm. Three data sets with different resolutions were generated to investigate the effect of varying resolution on permeability estimation. The first data set was created with a low resolution of 200 µm in all spatial directions. The dataset contains 578 × 1,962 × 42 voxels. The second data set was generated with a medium resolution of 100 µm in each spatial

direction. The dataset contains 1,154 × 3,922 × 84 voxels. In the third data set, the resolution was again reduced by half to 50 µm in all spatial directions. The dataset contains 2,308 × 7,844 × 169 voxels. The fracture was considered to be fully permeable with no low permeable filling or sealing. The matrix was due to previous studies considered to be fully impermeable (Cheng et al., 2020; Hassanzadegan et al., 2012). Thus, each image contains the permeable fracture in white colors (binary value = 1) and the impermeable matrix in black colors (binary value = 0).

Since HomCloud can only handle a maximum data volume of $1021^3$ voxels in a single analysis run, the data sets were divided into several subpackages, which were processed separately (Figure 2). The dataset with low resolution (200 µm/pixel) has 578 × 1,962 × 42 voxels and were split into 2 divisions in y-direction. Thus, 2 subpackages were created for the data set with low resolution, each of which contained 578 × 981 × 42 voxels. The x-direction and y-direction for the dataset were not necessary because the length of one side of the image was < 1021 pixels. The dataset with middle resolution (100 µm/pixel) has 1,154 ×

3,922 × 84 voxels and were split into 2 divisions in x-direction and 4 divisions in y-direction. The dataset was divided into 8 subpackages with an average of 577 × 981 × 84 voxels each. The dataset with high resolution (50 µm/pixel) has 1,154 × 3,922 × 84 voxels and were split into 3 divisions in x-direction and 8 divisions in y-direction. The data set was divided into 24 subpackages with an average of 769 × 981 × 169 voxels each.

PH analysis was then performed for each subpackage using Homcloud. There are different types of topological features (e.g.,

holes) in a 3D object. A connected component such as an impermeable, solid matrix is characterized as a 0-dimensional hole. Structures such as fractures or connected pores, which range from an inlet to an outlet of the domain are recognized as 1-dimensional (1D) holes. Lastly, there is also a 2-dimensional (2D) hole, which is represented by enclosed pores or fractures





not connecting to an inlet or outlet of the domain. Since only fractures connected between an inlet and an outlet of the domain can serve as potential flow channels, this study focuses on the analysis of 1D holes.

In HomCloud, a process called Filtration is used to detect 1-dimensional holes. For simplicity, an example of a two-dimensional image is shown in Figure 3. In the binary image, black area is a rock skeleton (binary value = 0), while white is void space (binary value = 1). The black pixels (binary value = 0) are thinned or thickened by one pixel from the boundary between white and black pixels. The process is considered as time variation. The initial image is assumed to be time 0 ($t = 0$). The time change in the negative direction is to make the black pixel thin, and the time change in the positive direction is to make the black pixel

thick. Continuing this operation, the image becomes all white if the pixels continue to be thinned. The images are saved for each time. As time passes from the all-white state to the positive direction, a channel (i.e., 1-dimensional hole) appears and disappears at certain times, respectively. Note that the channel (1-dimensional hole) here refers to a connected shape from left to right, surrounded by black pixels. These times are called birth ($b$) and death ($d$), respectively. In PH analysis, various holes are characterized using these birth-death pairs, which provide not only the topological features but also the geometric

information. Because it tries to see how persistent the hole is, it is called "persistent homology."

From the definitions of birth and death, the presence of a flow channel in an image means that birth-time is negative and death-time is positive. Thus, the number of such pairs ($b < 0 < d$) can be considered the number of flow channels in the image. In addition, in the case of the process of thickening the black area as shown in Figure 3, the fracture is closed from both sides (see the image "$t = d$"). Therefore, the doubling of death and multiplying by the resolution of the image can be taken as the

smallest aperture of the channel.

In this study, it is assumed that the channels are parallel-plate geometries and that they are parallel to each other. The permeability is estimated based on the power law (Suzuki et al., 2021):

$$K = \sum_{i=1}^{N} \frac{w_i h_i^3}{12A} \tag{1}$$

$A$ is the surface area of the cross section of the medium and $N$ is the number of flow channels. $w_i$ is the depth of flow channel and $h_i$ is the aperture of the flow channel $i$. The average of the depth of flow channel $\bar{w}$ is determined by the image (Suzuki et

al., 2021). The number of flow channels $N$ was estimated from the number of birth-death pairs and the aperture $h_i$ was estimated as $2d_i\delta$ in PH analysis. Thus, the above equation can be converted to the parameters from PH and image analysis.




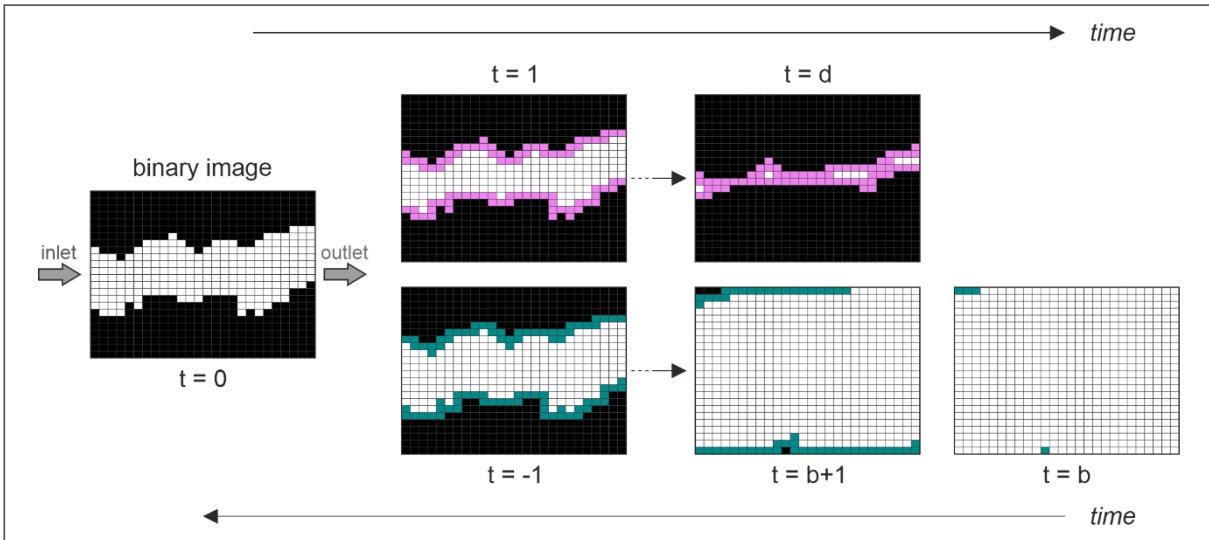

Figure 3: Schematic illustration of the filtration process during persistent homology analysis. The permeable area of the binary image (fracture) is thickened and thinned.

## 2.4 Experimental Measurement of the Fracture Permeability

An air permeameter was used to experimentally measure the fracture permeability. The transient air permeameter TinyPerm 3 (manufactured by New England Research Inc.) is a portable device, which is able to directly measure matrix permeabilities or hydraulic fracture apertures on outcrops or cores (Filomena et al., 2014; Cheng et al., 2020; Hale and Blum, 2022). For this purpose, the device is filled with air by lifting the piston and the rubber nozzle of the instrument is pressed against the fracture outlet. The measurement is started by depressing the piston in direction of the sample creating a vacuum between the fracture and the device. A microcontroller unit in the air permeameter simultaneously records the transient change in air pressure at the fracture outlet and the volume change in the device. Once the total volume of the device is compressed, the permeability is automatically determined from the recorded curve (Brown and Smith, 2013). The range of measurable permeabilities is from 1 mD to 10 D for porous rocks and hydraulic apertures of approximately 10 µm to 2 mm for fractures (New England Research Inc., 2016). The latter corresponds to permeabilities in orders of magnitude from $10^{-11}$ m² to $10^{-7}$ m².

This study uses measurements from Hale et al. (2022), which conducted experiments along the long edge (y-direction) of the fractured block and measured the permeability in x-direction. In addition, complementary measurements along the x-axis with permeabilities measured in y-direction were performed in the frame of this study. In total, the y-axis was divided into 21



sections, the x-axis into 4 sections. The corner areas were not included in the measurement due to breakouts from the block.

Each section was measured 10 times to obtain average values. The hydraulic aperture along the y-edge in x-direction was

determined to be 81 ± 1 µm (Hale and Blum, 2022), that along the x-edge in y-direction to be 57 ± 1 µm.

## 2.5    Numerical Flow Simulation

Despite the experimental air permeameter measurements, hydraulic apertures of the fracture were also determined by

numerical flow simulations using the Multiphysics Object Oriented Simulation Environment (MOOSE) framework (Permann

et al., 2020). Within this framework, the fluid flow through the fracture was simulated with the SaintBernard application

(Schädle, 2020), which is based on the inbuilt PorousFlow module (Wilkins et al., 2020; Wilkins et al., 2021). In this

application, a 3D fracture is projected on a 2D surface embedded in a 3D environment. The aperture of the fracture is assigned

as a permeability parameter to each cell of the 2D mesh. The fluid flow velocity is then simulated in the lower dimension and

the hydraulic aperture is calculated considering Darcy flow and the cubic law with the following equation for each cell of the

mesh:

$$a_h = \sqrt[2]{\frac{12 v \mu L}{\Delta p}} \tag{2}$$

Further information about the numerical simulation and SaintBernard can be found in Javanmard et al. (2021).

Similar to the air permeameter measurements, numerical flow simulations were performed with fluid flow in both, x- and y-

direction, to determine the permeability in each of these. These resulted in hydraulic apertures of 85 µm in x-direction

($6.0 \times 10^{-10}$ m²) and 73 µm in y-direction ($4.4 \times 10^{-10}$ m²), respectively.

## 210   3    Results and Discussion

### 3.1    Permeability Estimation from Persistent Homology Analysis

The prepared binary data sets were used to calculated the permeability of the fracture. The permeability was calculated in two

different flow directions, for flow parallel to the x-axis and parallel to the y-axis (Figure 4). The calculated permeabilities for

the 200 µm-resolution data set are $6.4 \times 10^{-10}$ m² in x-direction and $6.2 \times 10^{-10}$ m² in y-direction. Using the cubic law, the

calculated hydraulic apertures are 88 µm and 86 µm, respectively. For the data set with 100 µm resolution, the permeabilities





are $4.4 \times 10^{-10}$ m² in x-direction and $4.0 \times 10^{-10}$ m² in y-direction. This corresponds to hydraulic apertures of 73 µm in x-direction and 69 µm in y-direction. For 50 µm resolution data set, the permeability in x-direction is $7.0 \times 10^{-10}$ m² and $3.2 \times 10^{-10}$ m² in y-direction, which equals hydraulic apertures of 92 µm and 62 µm, respectively. Comparing the three different data set shows that PH analysis for all data sets result in higher permeability in x-direction than in y-direction ($k_x/k_y > 1.0$). A

look at the individual fracture surfaces, as well as the matched fracture, shows that the highest mechanical apertures of > 1 mm occur mainly along the barite vein that crosses the fracture parallel to the x-direction (Figure 4). From previous studies on the fracture, it appears that this barite vein dominates the flow behavior and, thus, forms the main flow path along the fracture due to its increased mechanical aperture and lower roughness compared to other regions of the fracture (Hale et al., 2020). Hence, it serves as a preferential flow path in x-direction, whereas it acts more as a barrier or redirection for flow in y-direction.

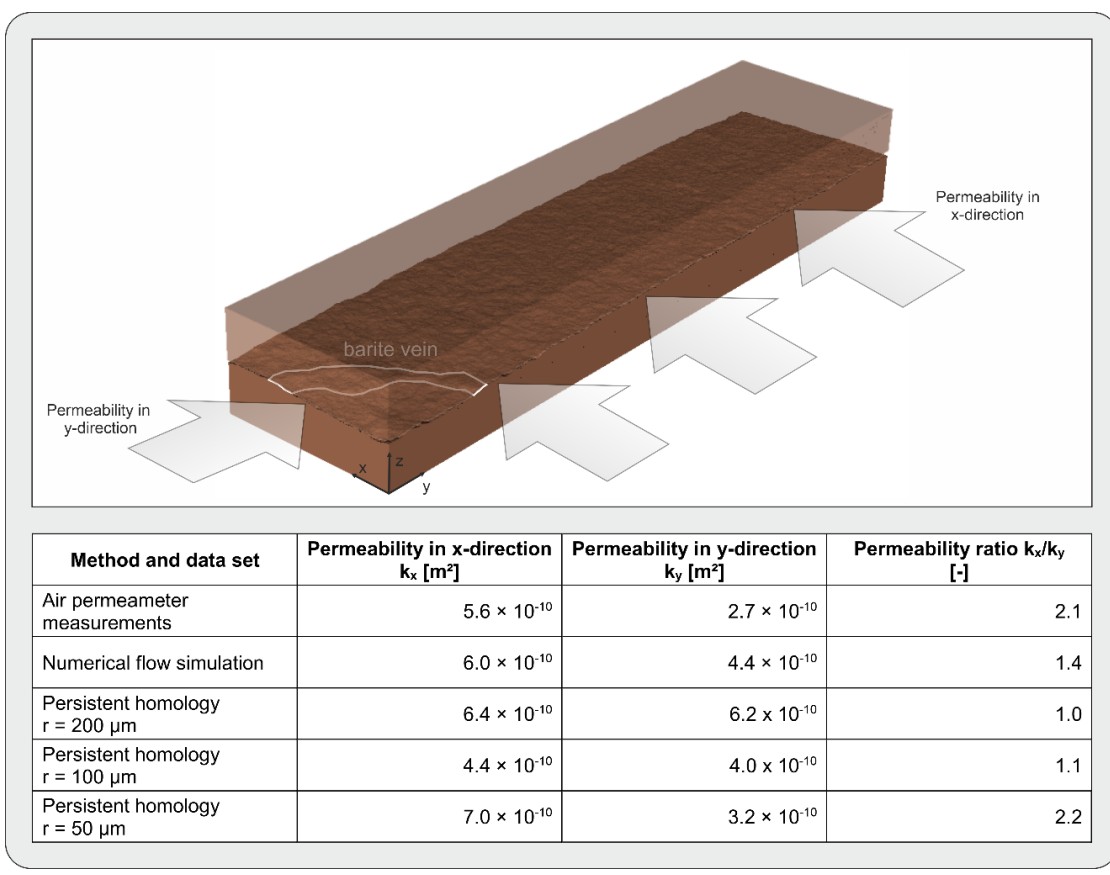

| Method and data set | Permeability in x-direction $k_x$ [m²] | Permeability in y-direction $k_y$ [m²] | Permeability ratio $k_x/k_y$ [-] |
|---|---|---|---|
| Air permeameter measurements | $5.6 \times 10^{-10}$ | $2.7 \times 10^{-10}$ | 2.1 |
| Numerical flow simulation | $6.0 \times 10^{-10}$ | $4.4 \times 10^{-10}$ | 1.4 |
| Persistent homology r = 200 µm | $6.4 \times 10^{-10}$ | $6.2 \times 10^{-10}$ | 1.0 |
| Persistent homology r = 100 µm | $4.4 \times 10^{-10}$ | $4.0 \times 10^{-10}$ | 1.1 |
| Persistent homology r = 50 µm | $7.0 \times 10^{-10}$ | $3.2 \times 10^{-10}$ | 2.2 |


Figure 4: Permeability determined by air permeameter measurements, numerical flow simulations and persistent homology using data sets with resolutions of 200 µm, 100 µm and 50 µm.





Nevertheless, the $k_x/k_y$-ratio of the data sets with 200 µm and 100 µm resolution is nearly identical, 1.0 and 1.1, respectively,
whereas the ratio of the 50 µm data set shows a higher ratio of 2.2. However, the absolute values display that all values are is
similar range. It can be seen that the permeability in the x-direction of the 50 µm resolved data set better matches the higher
permeability of the 200 µm data set compared to the 100 µm data set. On the other hand, the permeability in the y-direction
fits better to the lower permeability of the 100 µm data set.

## 3.2    Comparison to Air Permeameter Measurements and Numerical Simulation

In addition to the estimation of the fracture permeability by PH, the results were also compared to permeabilities derived from
alternative methods. Thus, a comparison was performed using experimental air permeameter measurements and numerical
flow simulations to show the validity of the PH analysis. In Figure 4, the values for the permeability in x- and y-direction as
well as the ratio of the two permeability values of air permeameter measurements and numerical simulation are also shown.
All x-permeabilities differ in $< 1.5 \times 10^{-10}$ m² from the experimental or numerical results. However, the permeabilities in y-
direction scatter slightly more. Overall, there is a good fit between PH analysis and alternative methods, which is reflected by
a root mean squared error (RMSE) of $1.5 \times 10^{-10}$ m². Normalization with the difference between maximum and minimum of
all observed permeabilities leads to a normalized root mean squared error (NRMSE) of 0.34. However, there is also the trend
that the values agree increasingly better with the comparative values as the resolution of the data set increases. For example,
the RMSE of x- and y-permeability between air permeameter/numerical simulation and 200 µm resolved data set is $2.0 \times 10^{-10}$
$^{-10}$ m². For the 100 µm data set, it reduces to $1.2 \times 10^{-10}$ m² and for the 50 µm data set, it is $1.1 \times 10^{-10}$ m². The NRMSE are 0.54
(200 µm), 0.38 (100 µm) and 0.25 (50 µm), respectively. This trend was also shown in studies using persistent homology in
porous media or fracture networks before (Moon et al., 2019; Suzuki et al., 2021). The study of Moon et al. (2019), in which
fluid flow through pore spaces of different digital sandstone and chalk samples was examined, could show particularly that the
number of excessively high outlier permeabilities can be prevented with higher resolution. Similar findings are shown in
Suzuki et al. (2021), in which also permeabilities that are significantly higher than the comparison simulation could be reduced
by improving the resolution. Conclusively, it can be stated that apart from the permeability estimation of the 50 µm data set in
x-direction, the results of the PH analysis improve with increasing resolution.





### 3.3 Evaluation of Persistent Homology Analysis

Since the results of three different methods for permeability determination are in good agreement (NRMSE = 0.34), a classification of the results in the context of other PH analysis was carried out. The study of Suzuki et al. (2021) applied PH on different data sets of porous and fractured rocks of previous studies (Andrew et al., 2014; Muljadi et al., 2016; Mehmani and Tchelepi, 2017). 16 datasets of porous media and 15 datasets of fracture networks were analyzed, each with PH and numerical flow simulation. In Figure 5, the results of this study as well as the results by Suzuki et al. (2021) are shown. Two

main findings can be derived from this comparison: (1) The values determined in this study are in the same range of permeability as the data sets investigated in the previous study and (2) in both studies, PH tends to slightly overestimate permeability, especially at low permeabilities $< 10^{-11}$ m$^2$. In this study, 67 % of the PH results are higher than the comparing methodology. In the previous study, even 90 % of the PH results overestimate numerical simulation. However, overestimation of the results in this study is only minor or in the same order of magnitude compared to the other results.

Of particular interest for this study are the permeabilities of fracture networks, which are displayed as dark gray diamonds in Figure 5, since they are also based on fractured instead of porous material. In general, it can be identified that permeabilities of fracture networks are distributed closer around the 1:1 line compared to porous media values (light gray crosses) in Figure 5. In addition, it is also not surprising that the results of this study fit those of fracture networks rather than porous rocks. Since the most values from fracture networks are results of the analysis of fracture networks with plane fracture surfaces in the study

of Suzuki et al. (2021), it is possible to estimate the influence of surface roughness as well. The rough single fracture studied here shows the same trend of permeabilities, which are all overestimated slightly, as the planar fracture networks addressed. This suggests only a minor influence of the roughness on the final result of the PH analysis. Furthermore, the cubic law, which is theoretically only valid for plane parallel fractures, seems to be also valid for rough single fracture such as a relatively smooth bedding plane joint of a sandstone. This is in large agreement with many other studies that have investigated the

influence of the application of cubic law on permeability of rough fractures (Witherspoon et al., 1980; Brush and Thomson, 2003; Konzuk and Kueper, 2004; Qian et al., 2011). Witherspoon et al. (1980) investigated on artificially induced fractures in granite, basalt and marble and showed that independent of flow direction or closing of fracture, the cubic law stays valid. This



general concept was proven by later studies, but with restrictions in terms of the maximum Reynolds number to be below 1

for synthetically created random single fractures (Brush and Thomson, 2003; Qian et al., 2011) and artificially induced

dolomite fractures (Konzuk and Kueper, 2004). All these studies also found an overestimation of flow through a single fracture

by cubic law compared to the Stokes equations. The large proportion of overestimated permeabilities by PH analysis can be

due to this.

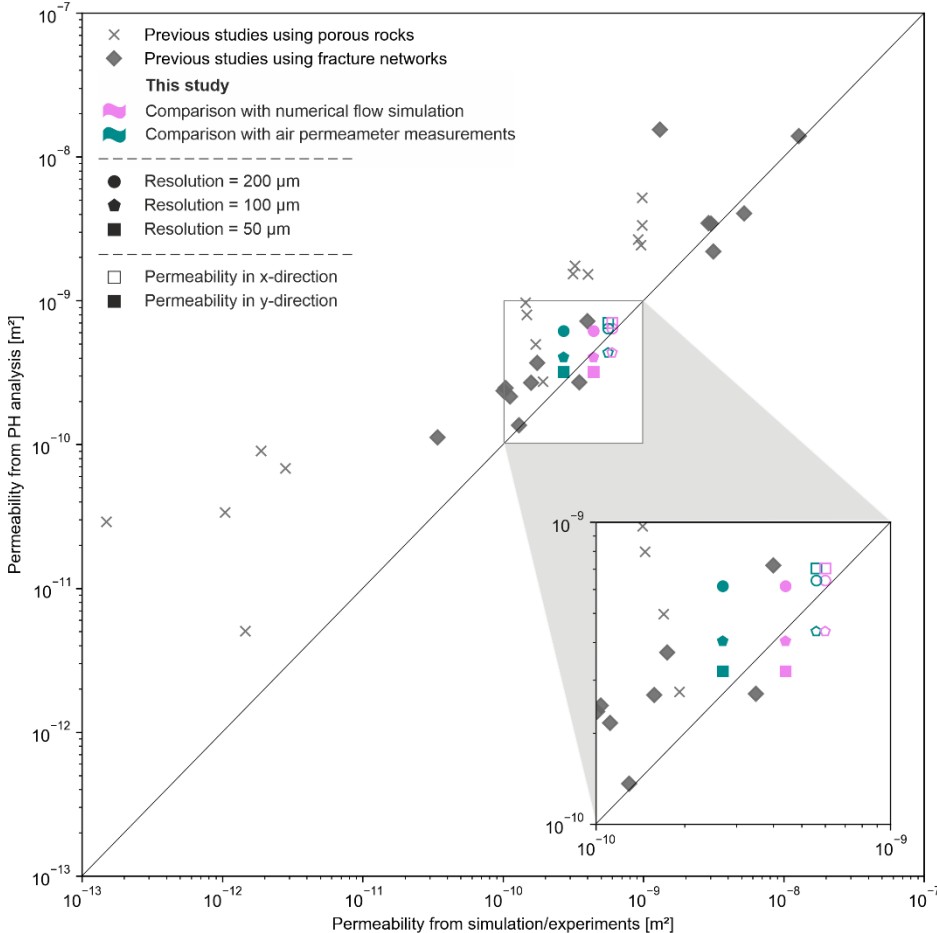

Figure 5: Comparison of the estimated permeabilities of this study with the estimated permeabilities for porous media (light

gray crosses) and fracture networks (dark gray diamonds) of previous studies by Suzuki et al. (2021) using data of Andrew et

al. (2014), Muljadi et al. (2016) and Mehmani and Tchelepi (2017)





Previously, it is shown that PH provides comparable results for permeability estimation of rough single fractures compared to other more conventional methods. However, for it to be an alternative, the effort and computational time also has to be

considered. In Table 1, an estimation of different working steps is presented for the permeability estimation of single fractures. For PH analysis, the dependency of the time needed on the resolution is also considered. The table is divided into three working steps. The step of preparation and preprocessing contains every working step after the collection of a sample including imaging and matching of fracture surfaces, generating binary images, setting up a numerical model or initializing the air permeameter. In the second working step, the effort for the actual measurement, numerical simulation or PH analysis is considered for 1

sample. In postprocessing, all working steps recalculating the fracture permeability from the measured, simulated or analyzed results is considered. Comparing the expenditure of time for all methods and resolutions, all methods apart from PH analysis for 50 µm data are ranging in the same order of magnitude. An increase to 50 µm resolution demands an enormous increase in time expenditure, mainly due to the extremely high analysis time. Considering the quality of the results, the data set with 100 µm therefore seems to be an adequate alternative to conventional methods, as it can provide high quality results with

similar efforts.

Table 1: Estimation of expenditure of time in hours for air permeameter measurements, numerical flow simulation and persistent homology for three different resolution steps (200 µm, 100 µm and 50 µm).

| Method | | Preparation and Preprocessing | Measurements/ simulation/ analysis | Postprocessing | Total |
|---|---|---|---|---|---|
| Air permeameter measurements | | 1.0 | 4.8 | 0.2 | 6.0 |
| Numerical flow simulation | | 7.0 | < 0.1 | 0.1 | 7.1 |
| Persistent homology analysis | 200 µm | 5.5 | 0.1 | < 0.1 | 5.6 |
| | 100 µm | 6.5 | 0.8 | < 0.1 | 7.3 |
| | 50 µm | 8.0 | 4.6 | 0.2 | 12.8 |

## 4    Conclusions

This study shows that persistent homology (PH) provides acceptable results for the permeability estimation of a natural bedding plane joint of a sandstone. This is particularly valid in the order of magnitude from $10^{-10}$ to $10^{-8}$ m$^2$. Compared to other methods such as experimental air permeameter measurements and numerical flow simulation, it tends to slightly overestimate lower permeabilities. However, the overestimation of permeabilities is also traceable in previous studies using PH analysis on porous

media as well as small-scale fracture networks. For single fracture application, a reason could be the application of the cubic law, which tends to overestimate the permeability in fracture networks or rough fractures.

In comparison to other methods, PH is a cheap and time-effective method. Once the geometry of a fracture is imaged (e.g. laser scanning, computed tomography, Structure from Motion), all the tools to determine permeability are open-source accessible. In contrast to most experimental methods, no laboratory is required to estimate the permeability. An exception to

this is the air permeameter used here, which is even applicable in field experiments due to its portability. The advantages of PH compared to numerical modeling are, firstly, the lower required computing capacity and computing time. Furthermore, the number of parameters required to successfully perform a simulation is significantly reduced, since only the geometry is sufficient as an input parameter.

Suzuki et al. (2020) showed that small-scale (millimeter to centimeter-scale) discrete fracture networks (DFN) can be precisely

studied by PH analysis. Our study demonstrates the applicability of the methods to a mesoscale (decimeter-scale), rough bedding joint of Flechtinger sandstone. In order to verify these results, future work should focus on other types of fractures, such as open mode or shear mode fractures, as well as different lithologies, such as fractures in granites or clay. By combining the approach of fracture networks and rough single fractures, a long-term objective could also investigate on fracture networks at larger scales under consideration of fracture surfaces roughness. In particular, the well-functioning subdivision of the total

data set into smaller, high-resolution subpackages should allow analysis of larger DFN without compromising resolution. However, other potential future work could be found in the comparison of such PH analysis with numerical DFN models. In addition, since numerical models for the accurate representation of fracture networks are computationally expensive, it is expected that PH is able to save time and costs.

**Keywords**

Topological data analysis; Persistent homology; Rough single fractures; Air permeameter measurements; Numerical flow simulation

**Acknowledgements**

Anna Suzuki was supported by JSPS KAKENHI Grant Number JP22H05108 (Japan) and JST ACT-X Grant Number JPMJAX190H (Japan). Furthermore, the authors like to thank Ruben Stemmle, Anna Albers and Hagen Steger for their support in conducting the air permeameter experiments and Keiichiro Goto for his support in performing the persistent homology analysis.

**Competing interests**

The authors declare that they have no conflict of interest.

**Author contribution**

M.F., A.S. and P.B. initiated the key concepts. M.F. prepared the data for persistent homology analysis, performed numerical simulations, conducted air permeameter measurements, and visualized the results. A.S. performed the persistent homology analysis and supervised the research. T.H. assisted the persistent homology analysis. P.B. supervised the research. M.F. wrote the original draft. All authors reviewed and edited the manuscript.

**Code/Data availability**

The binary image data of this study are available from the authors upon reasonable request.





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
