# Peer review of "Investigating rough single fracture permeabilities with persistent homology"

_EGUsphere, 2023_

## Author Comment (AC1)

Dear Editor, dear Reviewers,

we would like to thank you for your time and the constructive comments, which truly helped to improve the quality of the manuscript. Please find our detailed replies on the comments below. We hope that we answer all your remarks.

Our replies to the reviewer's comments are highlighted in blue. To highlight the nature of our replies we use a traffic light system indicating agreement with the reviewer marked in green, partial agreement in yellow, and objections in red.

Reviewers' and Editors' comments:

Reviewer #1:

The present article utilises persistent homology (PH) to estimate the permeability of a single fracture (12 by 45 cm length) in sandstone from high-resolution scans of the fracture surfaces. Results are compared to permeability estimations from numerical simulations and from measurements using an air permeameter. Results are interesting because they validate the use of PH as a method for permeability estimation. However, there are significant issues that I believe should be addressed before this work could be considered for publication.

  1.  The manuscript states that its goal is to address the influence of roughness on fracture permeability, but no parameters of the fracture roughness are provided. I strongly recommend including a roughness quantification for the fracture in the paper (e.g. roughness exponent from power spectral analysis).

We agree. There are already existing values for the Hurst exponent of the fracture surfaces, which were determined by Gutjahr et al. (2022). We therefore add this information as follows (L101-105): "Furthermore, the findings of Gutjahr et al. (2022), Hale et al. (2020) and Hale and Blum (2022) are crucial, since they performed investigations on fracture permeability on exactly the same fracture. Gutjahr et al. (2022) investigated on the roughness of the fracture and calculated the Hurst exponent for different angles. The medians of all Hurst exponents in x-direction and in y-direction are 0.48 and 0.42, respectively."

In addition, we discuss the Hurst exponent anisotropy briefly in context with the anisotropy of permeability (L245-249):
"The study of Gutjahr et al. (2022) shows that, in addition to the anisotropy of the permeabilities, a slight anisotropy of the roughness can be observed. Analogous to the permeability, the Hurst exponent in x-direction is higher (Hx = 0.48) than in y-direction (Hy = 0.42). It is noteworthy that the ratio of the Hurst exponents (Hx/Hy) is 1.1 and thus corresponds well with the determined ratios for permeabilities. This is reasonable, as increased roughness tends to result in more distinct flow channels with larger mechanical apertures. Consequently, this leads to increased permeability."

However, this study was designed to primarily investigate the general functionality of persistent homology for permeability estimation of a single fracture. This was to be investigated by examining the anisotropy in the x- and y-directions as well as the resolution dependency. We rephrase the sentences in our objective section (L88-90):
"The focus is on the anisotropy of permeability in different flow directions as well as the influence of

Karlsruhe Institute of Technology (KIT)
Kaiserstr. 12
76131 Karlsruhe, Germany
USt-IdNr. DE266749428

President: Prof. Dr.-Ing. Holger Hanselka
Vice Presidents: Prof. Dr. Thomas Hirth,
Prof. Dr. Oliver Kraft, Christine von Vangerow,
Prof. Dr. Alexander Wanner

LBBW/BW Bank
IBAN: DE44 6005 0101 7495 5001 49
BIC/SWIFT: SOLADEST600

resolution on permeability. Thus, three data set of the same fracture are prepared, which have different resolutions (50 µm, 100 µm and 200 µm)."

We also add a comment in out conclusions section to show that the actual influence of roughness is a crucial issue, which should be addressed in future work (L345-346):
"Furthermore, the influence of roughness on the flow behaviour and the permeability distribution across the fracture should be investigated."

2. In addition, the method of PH should be described more rigorously, and the discussion makes some questionable comparisons. Please find below my detailed comments.

We agree with the comment on the description of PH. Hence, we elaborated our description of persistent homology in section 2.3 as follows (L133-L194):

[revised manuscript text omitted]

==We partially agree== with the comment about questionable comparisons. The comparisons presented in this study are used to show that persistent homology is used for permeability assessment in single fractures in addition to porous media and fracture networks. In addition, the general trend is to be shown that persistent homology slightly overestimates the reference value. It is by no means intended to draw conclusions that single fractures in sandstone behave exactly as 3D-printed fracture networks.

3. I have made some language suggestions that the authors can decide whether to incorporate or not, but the manuscript would greatly benefit from thorough language editing.

==We agree== and implemented the language suggestions. A detailed list of the implemented changes can be found below. The line numbers refer to the revised manuscript and may differ slightly from the unrevised ones.

Abstract:
L11: the topological method **of** persistent homology
L12: the permeability **of** fracture networks
L13: **on** was deleted

Introduction:
L32: to study **how fluid flow is** influenced
L35: cheap**,**
L47-49: **rephrased**: Flow-through experiments allow to investigate direct fluid flow through fractures. The flow distribution and preferred flow paths can be predicted by replicating the fracture geometry in transparent materials.
L49: to determine **the** permeability
L50: **Air permeameters allow** to measure **the** permeability …
L51-53: rephrased: In addition, it is also possible to obtain a zonal observation of the permeability, since several measurements have to be conducted along an fracture outcrop due to the measurement method (Hale et al., 2020).
L63: **removed:** however
L66: without experiment**s** and numerical simulation**s.**
L69-70: **rephrased:** …, but adequate use of machine learning requires deep technical understanding, rigorous testing and sufficient amounts of training data.
L71: **from** big data
L72-73: **rephrased**: TDA is an analysis method that focuses on the structure of data within the field of algebraic topology, demonstrating particular strengths in handling data types such as images, complex structures, and networks.
L73-77: **rephrased**: TDA can capture the structure of the data in a rough sense and characterize the features of connectivity and holes, therefore, ignoring noises in data and extracting important information, independent of coordinate system and number of dimensions. Persistent homology (PH), one of the most used TDAs, can capture changes and continuity of topological features by tracking algebraic descriptions called homology.
L76: **rephrased:** most used (see comments of reviewer #2)
L77: **removed:** already
L79: …**, and** biology
L79-80: **rephrased:** In geosciences, this approach has only been applied in the past decade to characterize porous rocks and to determine their permeability
L84-85: **rephrased**: In these small-scale (millimeter to centimeter scale) studies the effect of fracture roughness was not particularly investigated

[Figure]

L88: **removed**: hence

Methods:
L96: with a **length** of 120 mm
L97-98: **rephrased:** Previous studies have already characterized relevant hydro-mechanical properties of Flechtinger sandstone such as …
L100-111: In fact, this is an incorrect statement of our side. The range of matrix permeability is rather 0.1-1 mD, which is based on the measured values of 0.17-0.36 mD by Hassanzadegan et al. (2012) and the statement of Cheng et al. (2020) that the permeability is in the range of $10^{-18}$ m². This was adjusted accordingly.

In addition, the following sentences were **rephrased**: Of particular interest for this study is the low matrix permeability of 0.1-1 mD (Cheng et al., 2020; Hassanzadegan et al., 2012). Furthermore, the findings of Gutjahr et al. (2022), Hale et al. (2020) and Hale and Blum (2022) are seminal, since they performed investigations on fracture permeability on exactly the same fracture. Gutjahr et al. (2022) investigated on the roughness of the fracture and calculated the Hurst exponent for different angles. The medians of all Hurst exponents in x-direction and in y-direction are 0.48 and 0.42, respectively. Hale et al. (2020) and Hale and Blum (2022) determined the average fracture permeability to be $5.6 \times 10^{-10}$ m². In addition, they found that the center of the fracture is less permeable than the left and right side of the fracture (according to the front view of the sandstone block shown in Figure 1a). On the right side of the fracture, this can be explained by a barite vein intersecting the fracture, which was formed before the fracture opening. In the closer vicinity of the vein, the mechanical aperture is increased compared to central parts of the fracture. Comparing fracture and matrix permeabilities shows that the fracture permeability exceeds matrix permeability by more than eight orders of magnitude. Thus, the matrix permeability is considered negligible in this study.
L128: … was then determined **by** applying …
L143-143: **rephrased**: Since the matrix permeability is eight orders of magnitude lower than the fracture permeability, the matrix was considered to be fully impermeable.
L187: … **have** parallel-plate geometries …

L209-211: We **rephrased** the sentence, since the reviewer misunderstood the statement: This study uses both, existing air permeameter measurements from Hale et al. (2020) and measurements conducted in this study. In the study of Hale et al. (2020) experiments along the long edge (y-direction) of the fractured block were performed and the permeability in x-direction was measured.
L217: **Apart from** experimental …
L228: **removed**: These resulted in hydraulic apertures of 85 µm in x-direction ($6.0 \times 10^{-10}$ m²) and 73 µm in y-direction ($4.4 \times 10^{-10}$ m²), respectively.

Results and discussion:
L241: **rephrased**: Detailed examination of the individual fracture surfaces and the matched fracture shows
L250: We agree that this is confusing and **removed** the following two sentences: It can be seen that the permeability in the x-direction of the 50 µm resolved data set better matches the higher permeability of the 200 µm data set compared to the 100 µm data set. On the other hand, the permeability in the y-direction fits better to the lower permeability of the 100 µm data set.
L283: **rephrased**: at relatively lower permeabilities
L290-291: **rephrased**: In addition, it is also not surprising that the results of this study have permeability values closer to those of fracture networks rather than porous rocks.
L294-295: **rephrased**: … the same trend of permeabilities, **the majority of which** are overestimated slightly, …

[Figure]
* * *
Reviewer #2:

Before presenting my comments, I would like to point out that my experience relevant for reviewing this manuscript is in topology-based methods for estimating fracture network connectivity, and geological analysis of fracture networks. Perhaps experience in geomechanical simulations of fracture networks is also useful. I am, however, no expert specifically in Persistent Homology (PH), or in laboratory or numerical flow simulations for fracture permeability estimation. Thus, for some of the questions or comments, an explanation might suffice.

Summary and general comments
The manuscript presents a novel topology-based method, Persistent Homology (PH), to estimate single fracture permeability from high-resolution imagery. The study demonstrates this application of PH using a sandstone sample and compares the resulting permeability estimates to values obtained with established methods for permeability measurements (air permeameter and numerical simulations). The main result of the study is that it validates the proposed use of PH, as it provides similar values for permeability as the established methods. The presented data and results are certainly interesting and of fair to good scientific value.

However, there are several issues in the presented manuscript that require major revisions before the publication of the study would seem justified. From my point of view, the main issues are:

1. There is a strong mismatch between the stated focus of the study and the presented results. The authors write that "The focus is on the influence of roughness of the fracture surfaces on the flow behavior and the determination of the permeability distribution across a natural bedding plane fracture." Yet, the manuscript contains no quantitative data on roughness or any other comparison of permeability between fractures of varying roughness that would allow the intended assessment. Permeability distribution is also limited to a very brief presentation of permeability anisotropy, which is also not satisfactory if this is the focus of the study.

We agree. Please also see our reply to comment #1 by Reviewer #1. This study was designed to investigate the general functionality of persistent homology for permeability estimation of a single fracture. This was to be investigated by examining the anisotropy in the x- and y-directions as well as the resolution dependency. We therefore rephrase the sentences in our objective section as follows (L88-90):
"The focus is on the anisotropy of permeability in different flow directions as well as the influence of resolution on permeability. Thus, three data set of the same fracture are prepared, which have different resolutions (50 μm, 100 μm and 200 μm)."

We also add a comment in out conclusions section to show that the actual influence of roughness is a crucial issue, which should be addressed in future work (L345-346):
"Furthermore, the influence of roughness on the flow behaviour and the permeability distribution across the fracture should be investigated further."

2. In part, the comparisons made with other studies in the results and discussion sections are difficult to justify, or they are not explained well enough. For instance, I had trouble seeing how absolute permeability values of a single fracture of a real sandstone can or should be

**www.kit.edu**

[Figure]

compared to permeabilities of fracture networks in an unspecified material (probably synthetic). The same applies to the comparison with trends of PH-derived permeability of porous media.

We partly agree. The comparisons presented in this study are used to show that persistent homology is used for permeability assessment in a single fracture in addition to porous media and discrete fracture networks. In addition, the general trend is to be shown that persistent homology slightly overestimates the reference value. It is by no means intended to draw conclusions that a single sandstone fractures (bedding plane) behaves exactly as 3D printed discrete fracture network (DFN). Nevertheless, we elaborated the section (L286-L296):

"Of particular interest for this study are the permeabilities of fracture networks, which are displayed as dark gray diamonds in Figure 5, since they are also based on fractured instead of porous material. In general, it can be identified that permeabilities of fracture networks are distributed closer around the 1:1 line compared to porous media values (light gray crosses) in Figure 5. In addition, it is also not surprising that the results of this study have permeability values closer to those of fracture networks rather than porous rocks. This is due to mechanical aperture of the individual fractures, which form a fracture network, being of a similar order of magnitude to the single fracture investigated here. Since the most values from fracture networks are results of the analysis of fracture networks with plane fracture surfaces in the study of Suzuki et al. (2021), it is possible to estimate the influence of surface roughness as well. The rough single fracture studied here shows the same trend of permeabilities, the majority of which are overestimated slightly, as the planar fracture networks addressed. This suggests only a minor influence of the roughness on the final result of the PH analysis. However, it should be considered that typically fracture surfaces have roughnesses of $H > 0.5$, whereas the roughness of the used fracture is slightly lower ($Hx = 0.48$ and $Hy = 0.42$)."

3. The authors conclusions on overestimation of permeability when using PH do not seem to convincingly match their own data. Note that is not necessarily a bad thing, because the presented permeability estimates match those from the other methods rather well.

We partly agree. The data shows that the experimentally or numerically determined reference p are slightly exceeded for the majority of the estimated permeabilities in this study (67 % of estimated permeabilities exceed their reference value). In fact, the overestimation of permeability is rather low compared to the other permeabilities presented. Since the same trend can be seen in the study of Suzuki et al. (2021), we have included the conclusion on overestimation of permeabilities in this study.

4. One interesting and perhaps critical feature in the sample, a barite vein, is qualitatively interpreted to have important influence on the fracture permeability. However, this feature is not mentioned in the sample description, nor is it described quantitatively.

We agree. Hence, the barite vein is now mentioned in the sample description as follows (section 2.1, L101-109):
"Furthermore, the findings of Hale et al. (2020) and Hale and Blum (2022) are crucial, since they performed investigations on fracture permeability on exactly the same fracture. They determined the average fracture permeability to be $5.6 \times 10^{-10}$ m². In addition, they found that the centre of the fracture is less permeable than the left and right side of the fracture (according to the front view of the sandstone block shown in Figure 1a). On the right side of the fracture, this can be explained by a barite vein intersecting the fracture, which was formed before the fracture opening. In the closer vicinity of the vein, the mechanical aperture is increased and the roughness is lower compared to central parts of the fracture."

[Figure]

5. Overall, I think that the rich high-resolution image data would allow for much more nuanced and quantitative analysis and discussion. The quality of the study would greatly benefit from including quantification of the fracture's properties (roughness, spatial variability) and combining that data with PH-derived permeability.

We partly agree. The primary aim of this study was to prove the general functionality of persistent homology for permeability estimation of single fractures. For this reason, this study is limited to the general trends and directionality of permeability. Certainly, a more detailed consideration of the permeability distribution in parts of the fracture would be possible based on the high-resolution scans. However, we see this as part of future work and therefore included the following paragraph in the "Conclusions" chapter (L347-349):
"In addition, a more detailed investigation of the permeabilities of different areas of this fracture could be performed on the basis of the high-resolution scans used here. This could also allow a potential scaling effect of the permeabilities to be analysed in more detail."

Regarding the inclusion of fracture properties such as roghness, please see our reply to comment #1 by Reviewer #1.

6. In addition to the scientific content, the authors might consider performing extensive language editing for conciseness and more clarity.

We agree and implemented the suggestions in our manuscript. The line numbers refer to the revised manuscript and may differ slightly from the unrevised ones:

Introduction:
L41-42: **rephrased:** Experimental methods provide more detailed results than empirical methods which use more simplified relations for fast practical application.
L59: **rephrased:** …, which would exceed technical possible conditions in laboratory experiments.
L61: **removed:** it has to be considered
L63: **removed:** however
L69-70: **rephrased:** …, but adequate use of machine learning requires deep technical understanding, rigorous testing and sufficient amounts of training data.
L71: **removed:** crucial
L76: **rephrased:** most used
L84-85: **rephrased**: In these small-scale (millimeter to centimeter scale) studies the effect of fracture roughness was not particularly investigated

Methods:
L101-111: In fact, this is an incorrect statement of our side. The range of matrix permeability is rather 0.1-1 mD, which is based on the measured values of 0.17-0.36 mD by Hassanzadegan et al. (2012) and the statement of Cheng et al. (2020) that the permeability is in the range of $10^{-18}$ m². This was adjusted accordingly.
In addition, the following sentences were **rephrased**: Of particular interest for this study is the low matrix permeability of 0.1-1 mD (Cheng et al., 2020; Hassanzadegan et al., 2012). Furthermore, the findings of Hale et al. (2020) and Hale and Blum (2022) are seminal, since they performed investigations on fracture permeability on exactly the same fracture. They determined the average fracture permeability to be 5.6 · 10-10 m². In addition, they found that the center of the fracture is less permeable than the left and right side of the fracture (according to the front view of the sandstone block shown in Figure 1a). On the right side of the fracture, this can be explained by a barite vein intersecting the fracture, which was formed before the fracture opening. In the closer vicinity of the vein,

the mechanical aperture is increased and the roughness is lower compared to central parts of the fracture. Comparing fracture and matrix permeabilities shows that the fracture permeability exceeds matrix permeability by more than eight orders of magnitude. Therefore, the matrix permeability is considered negligible in this study.

L142: The barite vein is not sealing the fracture, since it was created before the opening of the fracture. For clarification, we added this statement in L107-108.

L143-144: **rephrased**: Since the matrix permeability is eight orders of magnitude lower than the fracture permeability, the matrix was considered to be fully impermeable.

L164: black areas are **the rock matrix** …

L197: … is thickened (**cyan**) and thinned (**pink**)

L224-225: The following sentence was **added**: In equation 2, ah is the hydraulic aperture of the fracture, v is the fluid flow velocity, µ is the dynamic viscosity, L is the length of the fracture (in flow direction) and Δp is the hydraulic pressure gradient.

L228: Yes, the results were obtained by cubic law, but they were removed anyway (c.f comments to reviewer #1)

Results and discussion:

L242: We added the description of the barite vein in the methods (c.f. comment to L101-110)

L260: **removed**: Nevertheless

L285: … to the results of **Suzuki et al. (2021)**.

L29-2987: **rephrased**: Furthermore, the local cubic law, seems to be also valid for rough single fracture such as a bedding plane joint of a sandstone

L298-299: This is **overall in good agreement** with …

L299: **local** cubic law

L311: **In previous sections**, it is shown that …

L331: **removed**: This is particularly valid in the order of magnitude from $10^{-10}$ to $10^{-8}$ m².

L332: **removed**: However

L350: **removed**: However

---

## Referee Report (RR1)

Overall, the revised manuscript has improved significantly and addressed most points made by the reviewers. However, two significant comments (made by both reviewers) were ignored or addressed only very briefly, which I did not find satisfactory.

Note: I have copy-pasted the original comment (blue) and response (green) from the author's response document, and write my new comment below (yellow). Some parts of specific relevance are marked bold.

2. In part, the comparisons made with other studies in the results and discussion sections are difficult to justify, or they are not explained well enough. For instance, I had trouble seeing how absolute permeability values of a single fracture of a real sandstone can or should be compared to permeabilities of fracture networks in an unspecified material (probably synthetic). The same applies to the comparison with trends of PH-derived permeability of porous media.

We partly agree. The comparisons presented in this study are used to show that persistent homology is used for permeability assessment in a single fracture in addition to porous media and discrete fracture networks. In addition, the general trend is to be shown that persistent homology slightly overestimates the reference value. It is by no means intended to draw conclusions that a single sandstone fractures (bedding plane) behaves exactly as 3D printed discrete fracture network (DFN). Nevertheless, we elaborated the section (L286-L296): "Of particular interest for this study are the permeabilities of fracture networks, which are displayed as dark gray diamonds in Figure 5, since they are also based on fractured instead of porous material. In general, it can be identified that permeabilities of fracture networks are distributed closer around the 1:1 line compared to porous media values (light gray crosses) in Figure 5. **In addition, it is also not surprising that the results of this study have permeability values closer to those of fracture networks rather than porous rocks. This is due to mechanical aperture of the individual fractures, which form a fracture network, being of a similar order of magnitude to the single fracture investigated here.** Since the most values from fracture networks are results of the analysis of fracture networks with plane fracture surfaces in the study of Suzuki et al. (2021), it is possible to estimate the influence of surface roughness as well. **The rough single fracture studied here shows the same trend of permeabilities, the majority of which are overestimated slightly, as the planar fracture networks addressed.** This suggests only a minor influence of the roughness on the final result of the PH analysis. However, it should be considered that typically fracture surfaces have roughnesses of $H > 0.5$, whereas the roughness of the used fracture is slightly lower ($Hx = 0.48$ and $Hy = 0.42$)."

- I understand the intention, but here the comparisons remain questionable in my view:
  - The authors state it is not the goal to suggest that fracture networks and single fractures behave the same. Yet, they bring

up **absolute** permeability values for comparison to make the results seem plausible. Two "fractured media" are insufficient as grounds for such a comparison of absolute values. Matching absolute values may be coincidental, even though the mechanical aperture seems to be similar as stated. But I would actually be surprised if in that case a non-rough network and single frac result in the same k values.

o NOTE: the stated goal of the paper is to show the applicability of PH for single fracture permeability estimation. This has been shown nicely in the paper by comparing to other methods. I'd argue that this is the value of the present study. The comparison of the results to absolute values of other studies with different study objects is unnecessary and hard to justify. I srongly suggest to delete them.

- On the other hand, it is useful to compare trends for overestimation (if any, see comment below) and state that they seem in line with other studies. This is because one compares methods, not specific samples/media.

3. The authors conclusions on overestimation of permeability when using PH do not seem to convincingly match their own data. Note that is not necessarily a bad thing, because the presented permeability estimates match those from the other methods rather well.

NOTE: this is similar to the comments of reviewer 1 for L261 and L271 in the original submission. It seems like the authors chose to ignore these two comments entirely, which in itself is unsatisfactory.

We partly agree. The data shows that the experimentally or numerically determined reference p are slightly exceeded for the majority of the estimated permeabilities in this study (67 % of estimated permeabilities exceed their reference value). In fact, the overestimation of permeability is rather low compared to the other permeabilities presented. Since the same trend can be seen in the study of Suzuki et al. (2021), we have included the conclusion on overestimation of permeabilities in this study.

- I remain with my concern that the conclusion on overestimation seems rather forced given the presented data. I suggest to rephrase it (see below). Interestingly, even the authors themselves state that "the overestimation of permeability is rather low compared to the other permeabilities presented". Then why is overestimation presented as one of the conclusions, rather than for instance a "good match"?

- Specifically, the 67% (8 of 12 points) seem like a clear trend, but I think the presentation of only this number is a bit misleading. This is because the 12 points only represent 6 measurements, compared to two different methods at 3 different resolutions. Looking closely, it is clear that the conclusion on over-/underestimation is also resolution and direction-dependent. In my view, this is a strong indication that a general

statement in terms of over- or underestimation is not possible based on this data.
- As a hopefully constructive suggestion, I think the authors can conclude that differences to the results of other, established methods are small (=good match), and more data are needed to analyse the impact of resolution and anisotropy on the results. In this context, the results of Suzuki et al. (2021) can still be stated to give scientific context – but the presented study here does not convincingly confirm their results in my view.
-

---

## Author Response (AR2)

Dear Editor, dear Reviewer,

we would like to thank you again for your time and the constructive comments. Please find our detailed replies on the comments below. We hope that we answer all your remarks.

Our replies to the reviewer's comments are highlighted in blue. To guide you better through the reply letter, comments of reviewers and the corresponding answers of the authors made in the initial review, which are still relevant for our reply, are marked in *grey*. To highlight the nature of our replies we use a traffic light system indicating agreement with the reviewer marked in green, partial agreement in yellow, and objections in red.

Reviewers' and Editors' comments:

Reviewer #2:

1. *In part, the comparisons made with other studies in the results and discussion sections are difficult to justify, or they are not explained well enough. For instance, I had trouble seeing how absolute permeability values of a single fracture of a real sandstone can or should be compared to permeabilities of fracture networks in an unspecified material (probably synthetic). The same applies to the comparison with trends of PH-derived permeability of porous media.*

*We partly agree. The comparisons presented in this study are used to show that persistent homology is used for permeability assessment in a single fracture in addition to porous media and discrete fracture networks. In addition, the general trend is to be shown that persistent homology slightly overestimates the reference value. It is by no means intended to draw conclusions that a single sandstone fractures (bedding plane) behaves exactly as 3D printed discrete fracture network (DFN). Nevertheless, we elaborated the section (L286-L296):*
*"Of particular interest for this study are the permeabilities of fracture networks, which are displayed as dark gray diamonds in Figure 5, since they are also based on fractured instead of porous material. In general, it can be identified that permeabilities of fracture networks are distributed closer around the 1:1 line compared to porous media values (light gray crosses) in Figure 5. In addition, it is also not surprising that the results of this study have permeability values closer to those of fracture networks rather than porous rocks. This is due to mechanical aperture of the individual fractures, which form a fracture network, being of a similar order of magnitude to the single fracture investigated here. Since the most values from fracture networks are results of the analysis of fracture networks with plane fracture surfaces in the study of Suzuki et al. (2021), it is possible to estimate the influence of surface roughness as well. The rough single fracture studied here shows the same trend of permeabilities, the majority of which are overestimated slightly, as the planar fracture networks addressed. This suggests only a minor influence of the roughness on the final result of the PH analysis. However, it should be considered that typically fracture surfaces have roughnesses of $H > 0.5$, whereas the roughness of the used fracture is slightly lower ($Hx = 0.48$ and $Hy = 0.42$)."*

- I understand the intention, but here the comparisons remain questionable in my view:
  - The authors state it is not the goal to suggest that fracture networks and single fractures behave the same. Yet, they bring up absolute permeability values for comparison to make the results seem plausible. Two "fractured media" are insufficient as

Karlsruhe Institute of Technology (KIT)
Kaiserstr. 12
76131 Karlsruhe, Germany
USt-IdNr. DE266749428

President: Prof. Dr.-Ing. Holger Hanselka
Vice Presidents: Prof. Dr. Thomas Hirth,
Prof. Dr. Oliver Kraft, Christine von Vangerow,
Prof. Dr. Alexander Wanner

LBBW/BW Bank
IBAN: DE44 6005 0101 7495 5001 49
BIC/SWIFT: SOLADEST600

grounds for such a comparison of absolute values. Matching absolute values may be coincidental, even though the mechanical aperture seems to be similar as stated. But I would actually be surprised if in that case a non-rough network and single frac result in the same k values.

- o NOTE: the stated goal of the paper is to show the applicability of PH for single fracture permeability estimation. This has been shown nicely in the paper by comparing to other methods. I'd argue that this is the value of the present study. The comparison of the results to absolute values of other studies with different study objects is unnecessary and hard to justify. I strongly suggest to delete them.

- On the other hand, it is useful to compare trends for overestimation (if any, see comment below) and state that they seem in line with other studies. This is because one compares methods, not specific samples/media.

We agree and delete the comparison of absolute values between the different samples as well as the assessment to derive the influence of roughness in this single fracture. Therefore, the following parts of the manuscript are **removed** (L279-281 and L286-305 of the manuscript after the initial review):

"Two main findings can be derived from this comparison: (1) The values determined in this study are in the same range of permeability as the data sets investigated in the previous study…"

**and**

"Of particular interest for this study are the permeabilities of fracture networks, which are displayed as dark gray diamonds in Figure 5, since they are also based on fractured instead of porous material. In general, it can be identified that permeabilities of fracture networks are distributed closer around the 1:1 line compared to porous media values (light gray crosses) in Figure 5. In addition, it is also not surprising that the results of this study have permeability values closer to those of fracture networks rather than porous rocks. This is due to mechanical aperture of the individual fractures, which form a fracture network, being of a similar order of magnitude to the single fracture investigated here. Since the most values from fracture networks are results of the analysis of fracture networks with plane fracture surfaces in the study of Suzuki et al. (2021), it is possible to estimate the influence of surface roughness as well. The rough single fracture studied here shows the same trend of permeabilities, the majority of which are overestimated slightly, as the planar fracture networks addressed. This suggests only a minor influence of the roughness on the final result of the PH analysis. However, it should be considered that typically fracture surfaces have roughnesses of $H > 0.5$, whereas the roughness of the used fracture is slightly lower ($Hx = 0.48$ and $Hy = 0.42$). Furthermore, the local cubic law, seems to be also valid for rough single fracture such as a bedding plane joint of a sandstone. This is overall in good agreement with many other studies that have investigated the influence of the application of local cubic law on permeability of rough fractures (Witherspoon et al., 1980; Brush and Thomson, 2003; Konzuk and Kueper, 2004; Qian et al., 2011). Witherspoon et al. (1980) investigated on artificially induced fractures in granite, basalt and marble and showed that independent of flow direction or closing of fracture, the cubic law stays valid. This general concept was proven by later studies, but with restrictions in terms of the maximum Reynolds number to be below 1 for synthetically created random single fractures (Brush and Thomson, 2003; Qian et al., 2011) and artificially induced dolomite fractures (Konzuk and Kueper, 2004). All these studies also found an overestimation of flow through a single fracture by cubic law compared to the Stokes equations. The large proportion of overestimated permeabilities by PH analysis can be due to this."

[Figure]

Since we still think comparing trends between the data of this study and the study of Suzuki et al. (2021) to proof that the methods works for different kind of cavities, we **added** the following section (L279-288 of the updated manuscript):

"The method of PH for permeability estimation in both studies provides comparable results to the respective reference method. The results of this study are closely scattered around the 1:1 line and therefore match well with the results based on fracture networks (dark grey diamonds). It appears that the results of this study can be estimated even better than most of the data points of previous data sets, especially those generated from porous media (light grey crosses). In the latter, PH tends to overestimate the permeability, which cannot be confirmed for the data in this study. However, this study indicates that the quality of the permeability estimation is not only attributable to the type of cavity (pores, single fracture or fracture networks). Based on our results, the quality of the permeability estimation by PH is also dependent on the resolution and anisotropy of the respective data set. Nevertheless, a larger number of data sets should be examined for a more precise assessment of the various influences on the quality of the permeability estimation."

2. *The authors conclusions on overestimation of permeability when using PH do not seem to convincingly match their own data. Note that is not necessarily a bad thing, because the presented permeability estimates match those from the other methods rather well.*

*We partly agree. The data shows that the experimentally or numerically determined reference p are slightly exceeded for the majority of the estimated permeabilities in this study (67 % of estimated permeabilities exceed their reference value). In fact, the overestimation of permeability is rather low compared to the other permeabilities presented. Since the same trend can be seen in the study of Suzuki et al. (2021), we have included the conclusion on overestimation of permeabilities in this study.*

- I remain with my concern that the conclusion on overestimation seems rather forced given the presented data. I suggest to rephrase it (see below). Interestingly, even the authors themselves state that "the overestimation of permeability is rather low compared to the other permeabilities presented". Then why is overestimation presented as one of the conclusions, rather than for instance a "good match"?

- Specifically, the 67% (8 of 12 points) seem like a clear trend, but I think the presentation of only this number is a bit misleading. This is because the 12 points only represent 6 measurements, compared to two different methods at 3 different resolutions. Looking closely, it is clear that the conclusion on over-/underestimation is also resolution and direction- dependent. In my view, this is a strong indication that a general statement in terms of over- or underestimation is not possible based on this data.

- As a hopefully constructive suggestion, I think the authors can conclude that differences to the results of other, established methods are small (=good match), and more data are needed to analyse the impact of resolution and anisotropy on the results. In this context, the results of Suzuki et al. (2021) can still be stated to give scientific context – but the presented study here does not convincingly confirm their results in my view.

We also agree on this issue. In addition to the removed parts shown in reply to comment #1 of reviewer #2 in this letter, we also **remove** additional parts regarding the overestimation of our own data (L281-285 of the manuscript after the initial review):

[Figure]

"…and (2) in both studies, PH tends to slightly overestimate permeability, especially at relatively lower permeabilities $< 10^{-11}$ m$^2$. In this study, 67 % of the PH results are higher than the comparing methodology. In the previous study, even 90 % of the PH results overestimate numerical simulation. However, overestimation of the results in this study is only minor or in the same order of magnitude compared to the results of Suzuki et al (2021)."

As mentioned in reply to comment #1 of reviewer #2, we still show the results of Suzuki et al. (2021) for scientific content and to proof the methodology. This should be in good agreement to the comments of reviewer #2.
* * *
Editor:

1.  Dear authors,
    please note the re-review by Reviewer 2; and also that you did not address the later comments/suggestions of Reviewer 1 (e.g. from "Line 272:" onwards in their initial review). These are important points and need to be addressed before I can consider a decision for publication.

    Comment of Reviewer #1 from L272 on:
    *"I don't think that you can conclude that there is only a minor influence of roughness on the PH analysis only because your permeability results are on the same range as those from Suzuki et al. (2021). First, you don't provide roughness measurements of the fracture sample, and therefore there are no quantitative parameters that allow to assess whether this fracture is rough or smooth. In fact, you call it 'rough' in the title and in line 270, and then 'relatively smooth' in line 274. I strongly recommend including a roughness quantification for this fracture in the paper. I don't know if PH allows to determine a roughness estimation, but your high-resolution scans would allow you to, for example, follow a workflow such as that by Candela et al. 2012, and determine the roughness exponent H from a power spectral analysis. There are many works that have used this approach, which would allow you to assess comparatively how rough/smooth is this fracture.*

    We agree, but we answered this part of the comment already in review letter #1 (c.f. reply to comment #1 of reviewer #1) since the reviewer made two similar comments on the same issue that no roughness measurement is given.

    *Second, the permeability of fractures has been shown to have fall within a very wide range of magnitudes (e.g. Walsh 1981, Kranz et al. 1979, Iwai 1976, Nara et al. 2011, and many others). That your permeability results happen to fall within the same range as those from Suzuki et al. (2021) seems coincidental, considering that you are comparing different scales and flow paths (fracture networks in a 5 cm length sample vs a single fracture of 12 by 45 cm length). I therefore don't think that you can derive meaningful conclusions from this comparison.*

    We agree. Since we think that this point addresses the same issue as comment #1 of reviewer #2, please find our answer to this comment in our reply to comment #1 of reviewer #2 in this letter.

**www.kit.edu**

[Figure]

*I also don't understand why you discuss your results in the context of the Cubic Law assumption of parallel plates (L273-274), when PH analysis obviously considers the topography of the fracture surfaces."*

We agree. As we removed the entire discussion about the influence of roughness on the results of PH in this part of the manuscript, we also deleted the part including discussion about Cubic Law (c.f. our reply to comment #1 of reviewer #2).